# Early-stage health technology assessment of fractional flow reserve coronary computed tomography versus standard diagnostics in patients with stable chest pain in The Netherlands

Iris W. A. Boot[1]*, R. Nils Planken[2,3], Alexander W. den Hartog[4], Hubertus J. M. Vrijhoef[1]

1 Panaxea B.V., Den Bosch, The Netherlands, 2 Department of Radiology and Nuclear Medicine, Amsterdam University Medical Centers, Location AMC, Amsterdam, The Netherlands, 3 Amsterdam UMC, University of Amsterdam, Amsterdam Cardiovascular Sciences, Amsterdam, The Netherlands, 4 Department of Cardiology, Amsterdam UMC – Location AMC, University of Amsterdam, Amsterdam, The Netherlands

* iris.boot@panaxea.eu

**Data Availability Statement:** All relevant data are within the manuscript and its Supporting information files.

## Abstract

### Objectives

The aim of this early-stage Health Technology Assessment (HTA) was to assess the difference in healthcare costs and effects of fractional flow reserve derived from coronary computed tomography (FFRct) compared to standard diagnostics in patients with stable chest pain in The Netherlands.

### Methods

A decision-tree model was developed to assess the difference in total costs from the hospital perspective, probability of correct diagnoses, and risk of major adverse cardiovascular events at one year follow-up. One-way sensitivity analyses were conducted to determine the main drivers of the cost difference between the strategies. A threshold analysis on the added price of FFRct analysis (computational analysis only) was conducted.

### Results

The mean one-year costs were €2,680 per patient for FFRct and €2,915 per patient for standard diagnostics. The one-year probability of correct diagnoses was 0.78 and 0.61, and the probability of major adverse cardiovascular events was $1.92 \times 10^{-5}$ and 0.01, respectively. The probability and costs of revascularization and the specificity of coronary computed tomography angiography had the greatest effect on the difference in costs between the strategies. The added price of FFRct analysis should be below €935 per patient to be considered the least costly option.

**Funding:** This work was funded by Hemolens Diagnostics B.V. (https://hemolens.eu/). This company specializes in cardiac diagnostics and has created a non-invasive method of identifying atherosclerosis based on algorithms used in computational fluid dynamics. The funders had no role in study design, data collection and analysis, decision to publish, or preparation of the manuscript.

**Competing interests:** The authors have declared that no competing interests exist.

## Conclusions

The early-stage HTA findings suggest that FFRct may reduce total healthcare spending, probability of incorrect diagnoses, and major adverse cardiovascular events compared to current diagnostics for patients with stable chest pain in the Dutch healthcare setting over one year. Future cost-effectiveness studies should determine a value-based pricing for FFRct and quantify the economic value of the anticipated therapeutic impact.

## Introduction

Coronary artery disease (CAD) is the leading cause of morbidity and mortality worldwide [1]. Although The Netherlands is considered a low risk country in this respect, CAD is the leading cause of morbidity and ranks fifth in causes of mortality in this country [2–4]. Early diagnosis is key to avoid cardiac events and deaths [5].

Consistent with guideline recommendations, obstructive CAD is generally diagnosed using coronary computed tomography angiography (CCTA) [6]. However, if the CCTA shows coronary stenoses it is often unclear what the functional significance of the lesion on the blood flow is. Traditionally, invasive coronary angiography (ICA) (with or without fractional flow reserve (FFR) measurement) or functional imaging would follow to further clarify to degree of stenosis [6]. To prevent unnecessary additional or invasive diagnostic procedures, measuring FFR from the non-invasive CCTA (FFRct) has been developed. With data from the CCTA scan, FFRct assesses the impact of stenosis in the coronaries using imaging data and mathematical simulations of the blood flow. Because of its accuracy, it could prevent unnecessary and costly treatments, including revascularizations, and diagnostic procedures without adding the burden and adverse events of an invasive technique [7–9].

The National Institute for Health and Care Excellence (NICE) issued guidance in 2017, which recommends FFRct for patients with recent onset stable chest pain [10]. Based on their comprehensive literature review, the authors concluded that FFRct may be cost saving in the UK (£391 per patient) compared to standard diagnostics and may reduce the need for ICA and revascularisation. NICE attaches most importance to the PLATFORM study, a prospective, consecutive cohort study, which enrolled 584 symptomatic outpatients (204 planned for non-invasive testing and 380 planned for invasive testing) who had an intermediate likelihood of obstructive CAD from 11 European sites [8]. After one year of follow up, the study indicated a statistically significant reduction of 33% in health care expenditure using FFRct-guided invasive testing compared to standard ICA ($8,127 vs. $12,145; p < 0.0001), but non-significantly higher costs in the planned non-invasive FFRct compared to the usual care group ($3,049 vs. $2,579; p = 0.82). A study performed in the United States showed similar costs for the initial evaluation of chest pain using FFRct compared to standard care pathways (cost difference +7% (+$324), CI -12% to +26%, p = 0.49) [11]. A single-center study performed in China also indicated that FFRct strategy is associated with a lower 1-year risk of MACE when compared to CCTA (HR: 1.73; 95% CI: 1.01, 2.95; p = 0.04) [12].

Because of the intention of introducing FFRct to hospitals in The Netherlands, and since diagnostic pathways are not standardized across countries, we conducted an early-stage health technology assessment (HTA) comparing total healthcare costs and effects of FFRct to standard diagnostics over one year in patients with stable chest pain and an intermediate pre-test probability of obstructive CAD in The Netherlands. Favourable findings could justify further investments in research and development of the technology.

## Methods

### Model overview

A decision-tree model was developed that integrated data from published literature and information about unit costs to compare costs and clinical outcomes of FFRct compared to standard diagnostics. To collect data on costs and effects a literature search was performed in Pubmed using the following search terms: (((coronary artery disease[MeSH Terms]) AND (cardiac diagnostic method[MeSH Terms])) AND (("2015/01/01"[Date—Completion]: "3000"[Date—Completion]))) AND ((coronary angiography[MeSH Terms]) OR (fractional flow reserve, myocardial[MeSH Terms]) OR (computed tomography[MeSH Terms])). Articles were excluded if they were published before 2015, if they did focus on other coronary malignancies than stable chest pain, and if they were not designed as a meta-analysis, systematic review or randomized controlled trial. The model simulates different diagnostic trajectories, including non-invasive anatomical imaging (i.e., CCTA), non-invasive functional imaging (i.e., magnetic resonance imaging (MRI)), and ICA with visual assessment and/or combined with a FFR measurement over a period of one year. The FFRct diagnostic strategy added the computational FFR analysis to the initial CCTA for all patients. Model outcomes included total costs from the hospital perspective, probability of correct diagnoses, and risk of major adverse cardiovascular events (MACE), defined as all-cause mortality, myocardial infarction (MI), and unplanned hospitalization for chest pain leading to urgent revascularization [8]. Outcomes were not discounted. The model was constructed in TreeAge Pro (version 2020; Williamstown, MA: TreeAge Software, Inc.).

Patients with stable chest pain and an intermediate pre-test probability of obstructive CAD enter the model and received a CCTA (with or without FFR analysis depending on the diagnostic arm) (Fig 1). If the test findings were inconclusive, meaning the test provided insufficient information about the effect of the lesion on the blood flow (i.e., no positive or negative test result), patients will undergo non-invasive functional imaging (i.e., MRI). If those findings are also inconclusive, patients will undergo an ICA with visual assessment. Finally, when those test findings were inconclusive, patients received ICA with an FFR measurement, in line with the transmural agreements for patients with non-acute chest pain of the Dutch Association of Cardiologist [6, 13]. Patients who received a positive test result (either true or false) after a test were assumed to be treated with medications or a percutaneous coronary intervention (PCI) or coronary artery bypass grafting (CABG). Patients who received a negative test result (either true or false) receive no further treatment. Patients with a conclusive test result (positive or negative) were at risk of MACE.

### Data and assumptions

The model input parameters used in the base case analysis, sources, and ranges of values used in sensitivity analyses are presented in Table 1 and described below.

**Test characteristics.** The sensitivity and specificity of each test were derived from previous published literature (Table 1) [7, 9, 17]. If there was information on test characteristics from multiple independent studies, a generic-inverse variance weighted average method was used to calculate the weighted mean of the test characteristics [18]. A sensitivity and specificity of 1 for ICA with the FFR measurement was assumed. We obtained the probability of an inconclusive test result from a study, which showed that 4.3% of patients awaiting non-coronary cardiac surgery had a non-diagnostic CCTA scan due to motion artefacts, blooming artefacts from extensive calcification, and insufficient coronary attenuation [16]. In the base case analysis, due to a lack of data, we assumed the same probability of inconclusive test results for

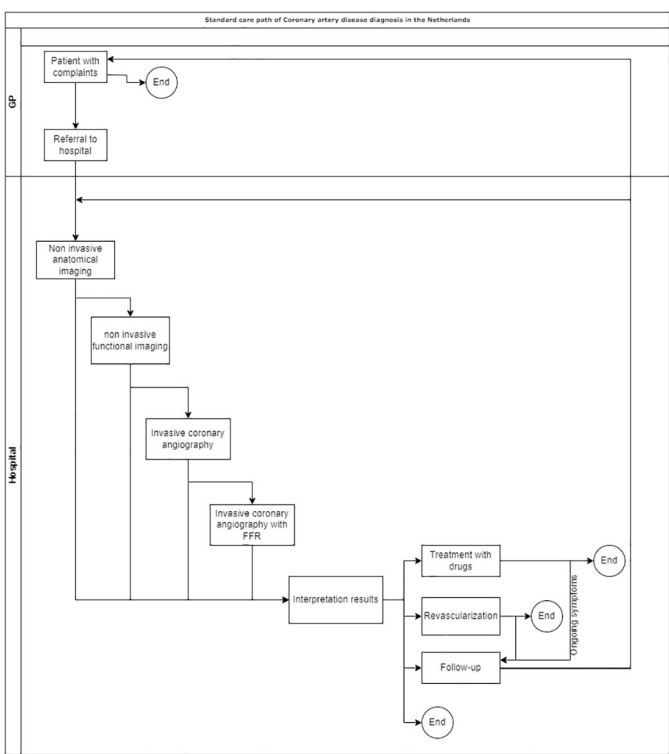

**Fig 1. Diagnostic care path for patients with stable chest pain in The Netherlands.** Abbreviations: GP: general practitioner; FFR: fractional flow reserve.

both strategies (i.e., CCTA in the FFRct and standard diagnostic arms) and for each screening option in the trajectory (i.e., MRI and ICA without FFR). We varied our assumptions in sensitivity analyses.

**Treatment efficacy.** The probability of MACE was obtained from the PLATFORM study [8]. The probability was dependent on the diagnostic strategy (FFRct or standard diagnostics) and whether the patients received invasive or non-invasive testing (Table 1).

The expected value of a correct diagnosis for each diagnostic strategy was calculated by including those patients who received a positive test result and had obstructive CAD (i.e., the true positives) and the patients who received a negative test result and did not have obstructive CAD (i.e., the true negatives).

**Costs.** Costs were calculated from the hospital perspective, including clinicians' time, disposables, equipment, medications, and treatments. Mean costs of the diagnostic procedures and treatments were calculated by multiplying the resource quantities with Dutch unit costs (S1 Table). The resource quantities were obtained from interviews with experts (i.e., 5 cardiologists and radiologists) from different hospitals and the unit costs from different published sources [20–22]. Experts were co-/authors identified via scientific publications or national guidelines. They were approached via email in December 2021 and the interviews were held in January and February 2022. All experts invited for the interviews provided written consent via email.

The costs of MACE were obtained by taking an average of the costs due to MI and costs due to CAD deaths obtained from published data [15, 19]. To calculate annual costs of the use of medications for patients with CAD, we adopted the Dutch National Health Care Institute guidelines for (maintenance) treatment of patients with stable angina pectoris [13]. A non-

**Table 1. Model input parameters and value ranges for sensitivity analyses.**

| Variables | Base case value | Range | Source |
|---|---|---|---|
| Pre-test probability of obstructive CAD | 0.317 | 0.153–0.492 | [14] |
| Probability of MACE* | | | |
| After invasive test (FFRct) | 0.010 | 0.001–0.037 | [8] |
| After non-invasive test (FFRct) | 0 | 0–0.035 | [8] |
| After invasive test (Usual care) | 0.011 | 0.001–0.038 | [8] |
| After non-invasive test (Usual care) | 0.010 | 0.0003–0.055 | [8] |
| Probability of treatment after + test result | | | |
| Medications (FFRct) | 0.484 | 0.387–0.581 | [15] |
| PCI (FFRct) | 0.491 | 0.393–0.589 | [15] |
| CABG (FFRct) | 0.025 | 0.02–0.03 | [15] |
| Medications (Usual care) | 0.345 | 0.276–0.414 | [15] |
| PCI (Usual care) | 0.635 | 0.508–0.762 | [15] |
| CABG (Usual care) | 0.021 | 0.017–0.025 | [15] |
| Uncertain test results | 0.043 | 0–0.1 | [16] |
| **Test characteristics** | | | |
| FFRct | | | |
| Sensitivity | 0.94 | 0.90–0.97 | [7, 9, 17, 18] |
| Specificity | 0.70 | 0.63–0.724 | [7, 9, 17, 18] |
| CCTA | | | |
| Sensitivity | 0.90 | 0.89–0.94 | [7, 9, 17, 18] |
| Specificity | 0.45 | 0.39–0.67 | [7, 9, 17, 18] |
| MRI scan | | | |
| Sensitivity | 0.90 | 0.75–0.97 | [7] |
| Specificity | 0.94 | 0.79–0.99 | [7] |
| ICA | | | |
| Sensitivity | 0.69 | 0.65–0.75 | [7] |
| Specificity | 0.67 | 0.63–0.71 | [7] |
| FFR ICA | | | |
| Sensitivity | 1 | 0.8–1 | assumption |
| Specificity | 1 | 0.8–1 | assumption |
| **Costs in € (2022)** | | | |
| CCTA (procedure)*** | 519 | 414–621 | [S1 Table] |
| MRI (procedure)*** | 617 | 494–740 | [S1 Table] |
| ICA (procedure)*** | 2175 | 1237–1856 | [S1 Table] |
| FFR ICA (procedure)*** | 820 (+ICA) | 1841–2761 | [S1 Table] |
| FFRct (procedure)*** | 700 (+CCTA) | 560–840 | [S1 Table] |
| MACE (event) | 6287 | 5030–7545 | [14, 15] |
| Medication mild CAD (annual)*** | 28 | 23–34 | [S2 Table] |
| Medication severe CAD (annual)*** | 139 | 111–167 | [S2 Table] |
| PCI (procedure) | 5105 | 4084–6126 | [19] |
| CABG (procedure) | 14560 | 11648–17472 | [19] |

Abbreviations: CAD: coronary artery disease; MACE: major adverse cardiovascular events, FFRct: fractional flow reserve from coronary computed tomography; PCI: percutaneous coronary intervention; CABG: coronary artery bypass grafting; CCTA: coronary computed tomography angiography; MRI: magnetic resonance imaging; = ICA: invasive coronary angiography; FFR: fractional flow reserve

*: defined as all-cause mortality, MI, and unplanned hospitalization for chest pain leading to urgent revascularization

**: See Supporting information, S1 File for detailed information

***: See Supporting information, S1 and S2 Tables for detailed information

drug treatment is suggested first. Thereafter several different drugs (i.e., acetylsalicylic acid, clopidogrel, bisoprolol, metoprolol, amlodipine, or felodipine) can be considered, between which can be switched. If the efficacy is still insufficient, possible combinations of the afore-mentioned drugs and isosorbide mononitrate and isosorbide nitrate in addition [13]. In the model for this study, it was assumed that people who undergo non-invasive tests and receive a positive result either take one or no drug, and people who undergo invasive testing and receive a positive result use a combination of drugs (or PCI or CABG). We multiplied the daily dos-ages by the unit costs of each type of medication and multiplied this by 365 to obtain annual total medication cost (S2 Table). We did not consider the costs of adverse events due to medications.

Information about the price of the FFR analysis (computational analysis only, excluding CCTA scan) was obtained from a Dutch distributor and amount to €700. All costs were trans-formed to 2022 prices using the price indices of the Dutch Healthcare Authority [23].

### Sensitivity analyses

One-way sensitivity analyses were conducted to determine the main cost drivers. The lower and upper values of the ranges used in the analyses correspond to the 95% confidence intervals reported by a single study or the highest and lowest values found in the published literature across multiple studies. For the cost parameters and the probabilities of medication, PCI, or CABG after a positive test result, a range of -20% to +20% of the baseline value was used. To determine the added price of FFRct analysis (computational analysis only, excluding CT scan) at which point both strategies are equal in costs, a threshold analysis was conducted.

## Results

### Literature review

The literature review resulted in 197 results. After title and abstract screening, 162 articles were excluded. After full text review, twelve articles were selected for inclusion, see Fig 2.

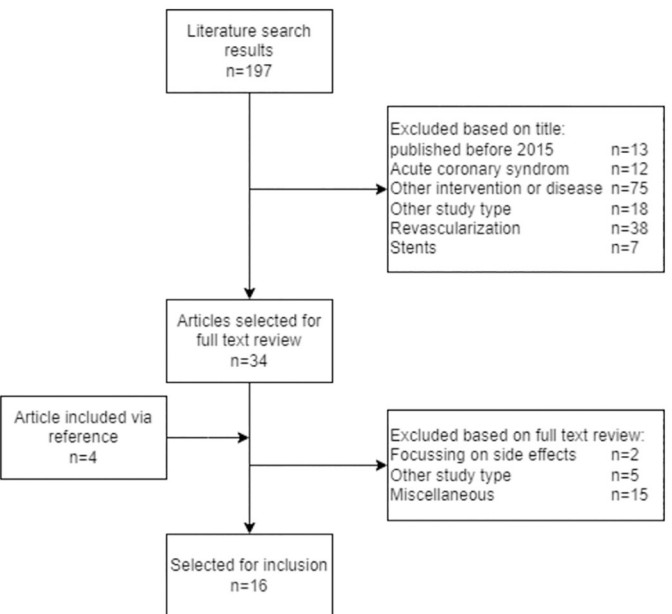

**Fig 2. Flow chart of literature review.**

## One-year health and costs outcomes

The mean total one-year costs were €2,680 per patient for FFRct and €2,915 per patient for standard diagnostics from the hospital perspective. FFRct has the lowest cost with a difference between the strategies of -€235 per patient. The one-year probability of correct diagnoses was 0.78 and 0.61, respectively. The one-year probability of MACE was $1.92 \times 10^{-5}$ for FFRct and 0.01 for standard diagnostics.

## Sensitivity analyses

Fig 3 shows the model input parameters that have the largest impact (indicated at the top) on the difference in costs between FFRct and standard diagnostics. These include the probabilities and cost of PCI and the specificity of CCTA. The least costly option switched from FFRct to standard diagnostics when the specificity of CCTA and the probability of PCI in the FFRct arm increased, and when the probability of PCI decreased in the standard diagnostics arm over their specified ranges.

The threshold analysis indicated that the added price of FFRct analysis should be below €935 per procedure to be considered the least costly option.

## Discussion

The objective of this early-stage HTA was to compare the costs and health benefits over one year of FFRct diagnostics to standard diagnostics in patients with stable chest pain with an intermediate pre-test probability of obstructive CAD in The Netherlands.

The results showed that FFRct may lead to reductions in total healthcare spending (-€235 per patient) and the risk of MACE ($1.92 \times 10^{-5}$ versus 0.01) and may increase the probability of correct diagnoses (0.78 versus 0.61). The least costly option switched from FFRct to standard diagnostics when the specificity of CCTA and the probability of PCI in the FFRct arm increased, and when the probability of PCI decreased in the standard diagnostics arm over their specified ranges. Moreover, the added price of FFRct analysis should be below €935 per procedure to be considered the least costly option.

The study findings are consistent with the NICE recommendations which indicate that FFRct-guided diagnostics may lead to cost savings [10]. Also, results from the PRECISE randomized trial endorse the results of more correct diagnoses by showing that more accurate catheterizations are performed with FFRct testing, without increasing the risk of MACE [24]. At last, additionally an increased catheterization lab efficiency and a reduction in additional non-invasive testing was shown in England in the FISH&CHIPS study [25]. However, individual trials do suggest, albeit non-significantly, higher costs in the FFRct-guided strategy compared to the usual care after planned non-invasive testing [8, 26]. In the FORECAST trial, for example, a multi-centre trial with 1,400 patients with stable chest pain were randomized to either CCTA with selective FFRct or standard care, which were dependent on their pre-test probability of CAD [26], showed a non-significantly higher costs in the FFRct-guided strategy compared to the usual care group at 9 months (£1,605 versus £1,491; p = 0.10). Differences in results can be explained by differences in standard diagnostic pathways between countries.

This study focused on people with an intermediate pre-test probability of obstructive CAD. The expert interviews clarified that the pre-test probability influences the choice of diagnostic path. People with intermediate pre-test probability are usually first referred to an CCTA scan. It is expected that the FFRct analysis can have the most benefit in this pathway and hence will be used for these patients. People with a higher pre-test probability are more likely to be directly referred to invasive tests. In this case we would expect the differences in costs to increase in favour of FFRct guided diagnostics, the amount of correct diagnosis to increase

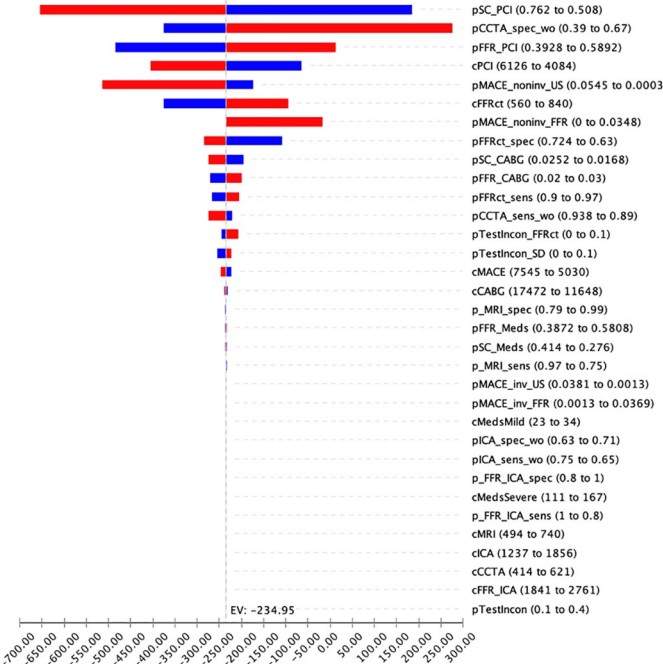

**Fig 3. Tornado diagram of the one-way sensitivity analyses.** This figure shows the model input parameters that have the greatest impact (indicated at the top) on the difference in costs between FFRct and standard diagnostics using a tornado diagram. The vertical axis indicates the names of the variables with the upper and lower values of the range used in the analysis in parentheses. The horizontal axis indicates the incremental (or difference in) cost between the two strategies. The dotted line indicates the expected value of the difference in cost using the base case values of the parameters. Using the base case values, FFRct is less expensive (-€234.95) compared to standard diagnostics. The red bars to the left of the dotted line indicate that when the value of that parameter increases, the difference in cost also increases (i.e., FFRct becomes more favourable); the blue bars to the left of the dotted line indicate that the difference in cost between the strategies will increase with decreasing values of that parameter. For example, if the specificity of CCTA decreases from 67% to 39%, the difference in costs will increase in favour of the FFRct strategy. Abbreviations: pSC_PCI: probability of percutaneous coronary intervention (PCI) in standard diagnostics arm; pCCTA_spec_wo: specificity of coronary computed tomography angiography (CCTA); pFFR_PCI: probability of PCI in the fractional flow reserve from coronary computed tomography (FFRct) arm; pMACE_noninv_US: probability of major adverse cardiovascular events (MACE) after a non-invasive test in the standard diagnostics arm; cPCI: costs of PCI; cFFRct: cost of FFR analysis (baseline: €700 per analysis); pMACE_noninv_FFR: probability of MACE after a non-invasive test in the FFRct arm; pTestIncon: probability of an inconclusive test result; pFFRct_spec: specificity of FFRct; pSC_CABG: probability of coronary artery bypass grafting (CABG) in the standard diagnostics arm; p_FFR_CABG: probability of CABG in the FFRct arm; p_FFRct_sens: sensitivity of FFRct; pCCTA_sens_wo: sensitivity of CCTA; cMACE: costs of MACE; pMACE_inv_US: probability of MACE after an invasive test in the standard diagnostics arm; pMACE_inv_FFR: probability of MACE after an invasive test in the FFRct arm; p_MRI_sens: sensitivity of magnetic resonance imaging (MRI); cCABG: costs of CABG; pFFR_Meds: probability of medications in the FFRct arm; pSC_Meds: probability of medications in the standard diagnostics arm; pICA_spec_wo: specificity of invasive coronary angiography (ICA); p_FFR_ICA_spec: specificity of fractional flow reserve from invasive coronary angiography; pICA_sens_wo: sensitivity of ICA; p_MRI_spec: specificity of MRI; p_FFR_ICA_sens: sensitivity of fractional flow reserve from invasive coronary angiography; cMedsMild: costs of medications for mild obstructive coronary artery disease patients; cMedsSevere: costs for severe obstructive coronary artery disease (CAD) patients; cCCTA: costs of CCTA; cFFR_ICA: costs of fractional flow reserve from ICA; cICA: costs of ICA; cMRI: costs of MRI.

and the risk of MACE to decrease. People with a lower pre-test probability usually first receive behavioural change advices. If this is not effective, they will usually undergo diagnostic imaging, starting from CCTA. Therefore, no differences in outcomes are expected compared to people with an intermediate pre-test probability when people are referred to imaging. For conclusive findings regarding costs, additional analyses could be performed to compare different pathways to FFRct guided diagnostics. Since FFRct adds a mathematical simulation to the

traditional CCTA, thereby increasing the accuracy of the diagnostic test, it is always expected to lower the amount of MACE.

FFRct has not been implemented in The Netherlands as standard care. Favourable findings for FFRct could justify further investments in research and development of the technology. For example, an algorithm that predicts the impact of blockages on the blood flow could be developed and optimized using artificial intelligence (AI). With deep learning techniques, a form of AI, the algorithm continues to predict more accurately by incorporating information and results from previous CCTA analyses.

The strength of the current study is the collection of Dutch unit cost data to explore the potential impact of the FFRct technology in the Dutch health care setting. The study has several limitations, which come with the territory of early-stage HTA and modeling. First, because the new technology has not been implemented yet we used data on test characteristics from published FFRct diagnostic studies. Although the techniques are similar, we plan to rerun the model with specific data when trial data becomes available. In our study a few assumptions were made. First, it was assumed that the diagnostic accuracy of ICA with FFR is one. If this accuracy would be lower in reality, this is expected to increase the risk of MACE in both treatment arms, but more in the standard of care arm since more people undergo invasive testing in that arm. Second, assumptions were made regarding the percentage of inconclusive test results. If this percentage would be higher in reality, both treatment arms would perform more diagnostic tests, both invasive and non-invasive. The opposite if the actual percentage would be lower. The first is expected to result in a greater difference between the two strategies, and the second in a smaller difference. Third, assumptions were made regarding drug prescription. The results of the sensitivity analyses show that costs of medications only have a small impact on the outcomes, see Fig 3. Besides, a difference in costs between strategies should always be considered in the light of a gain in health benefits. In this study, we focused only on the difference in costs from the hospital perspective between the strategies, but when more data becomes available a cost-effectiveness study and incorporating a broader (i.e., societal) perspective are warranted. In that case, costs such as productivity losses will also be included. These are expected to be in favour of the FFRct guided diagnostic strategy, since this strategy involves less invasive procedures and therefore less time in the hospital. This effect would be relevant in a working population. Next, travel time and costs would be included which are also expected to be lower for the FFRct guided strategy, since people have to visit the hospital fewer times. Besides, it is noteworthy that the specificity of MRI is higher than FFRct, see Table 1. Although it is expected that the combination of FFRct with CCTA would significantly improve diagnostic specificity [7], future studies could also give more clarity. A randomized controlled study (n = 528) evaluating the cost-effectiveness of adding advanced FFRct assessment to CT scans of coronary arteries is currently ongoing (FUSION study) [27]. Once the findings of this study become available, we plan to update the current model input parameters to investigate the impact on the findings. Third, we did not quantify the decision uncertainty or were able to direct future research priorities with value of information analyses. Instead, we showed the variables that had the greatest impact on the cost differences between the strategies and could potentially switch the least costly option. Finally, we did not incorporate that FFRct analysis can be targeted to a specific patient group, neither did we model that the test result may indicate whether an additional (invasive) test is necessary. In each case, we believe that this will improve the favourability of the FFRct diagnostic strategy. For hospitals diagnosing patients with stable chest pain in The Netherlands, the results of this study indicate that the implementation of FFRct guided diagnostics would decrease the costs per patient, the amount of adverse events and increase the probability of correct diagnoses. For hospitals in the Netherlands considering to implement FFRct, this study shows promising results. Moreover, these study

findings may contribute to the decision making process by the Dutch National Health Care Institute about reimbursement of FFRct as part of the standard health care package [28]. The authors would like to emphasize that this study focused on the healthcare setting in The Netherlands, therefore other settings require an analysis tailored to those specific settings.

Despite these limitations, we conclude that these early-stage HTA findings suggest that FFRct diagnostics may reduce total healthcare spending, probability of incorrect diagnoses, and MACE compared to current diagnostics for patients with stable chest pain in the Dutch healthcare setting. Future cost-effectiveness studies should determine a value-based pricing for FFRct and quantify the economic value of the anticipated therapeutic impact. Besides, if FFRct would be implemented, a longitudinal study is suggested to collect additional data on costs, diagnostic accuracy, and risk of MACE. Future studies could also focus on the broader healthcare system impact, including the impact of the implementation of FFRct guided diagnostics on healthcare system efficiency.

## Supporting information

**S1 Table. Dutch unit costs, resource quantities, and total costs of the diagnostic procedures.** Abbreviations: ICA: invasive coronary angiography; OR: operating room; FFR: fractional flow reserve; MRI: magnetic resonance imaging; IV: intravenous; CCTA: coronary computed tomography angiography; FFRct: fractional flow reserve from coronary computed tomography.
(DOCX)

**S2 Table. Mean costs of medications for patients with stable angina pectoris.** *The steps of administering medications are described in the guidelines for (maintenance) treatment for patients with stable angina pectoris by the National Health Care Institute in The Netherlands [13].
(DOCX)

**S1 File. Calculations of sensitivity and specificity per diagnostic tool based on the generic inverse variance weighted average [13].**
(DOCX)

## Acknowledgments

We acknowledge all experts who participated in the interviews.

## Author Contributions

**Conceptualization:** Iris W. A. Boot, Hubertus J. M. Vrijhoef.

**Data curation:** Iris W. A. Boot, Hubertus J. M. Vrijhoef.

**Formal analysis:** Iris W. A. Boot.

**Investigation:** Iris W. A. Boot.

**Methodology:** Iris W. A. Boot, Hubertus J. M. Vrijhoef.

**Project administration:** Iris W. A. Boot, Hubertus J. M. Vrijhoef.

**Supervision:** Hubertus J. M. Vrijhoef.

**Writing – original draft:** Iris W. A. Boot.

**Writing – review & editing:** Iris W. A. Boot, R. Nils Planken, Alexander W. den Hartog, Hubertus J. M. Vrijhoef.

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
