## [Decision Letter · Decision Letter 0]

22 Jan 2024

PONE-D-23-42875Early-stage health technology assessment of fractional flow reserve coronary computed tomography versus standard diagnostics in patients with stable chest pain in The NetherlandsPLOS ONE

Dear Dr. Boot,

Thank you for submitting your manuscript to PLOS ONE. After careful consideration, we feel that it has merit but does not fully meet PLOS ONE’s publication criteria as it currently stands. Therefore, we invite you to submit a revised version of the manuscript that addresses the points raised during the review process.

We look forward to receiving your revised manuscript.

Kind regards,

Haipeng Liu

Academic Editor

PLOS ONE

Journal Requirements:

3. Please be informed that funding information should not appear in the Acknowledgments section or other areas of your manuscript. We will only publish funding information present in the Funding Statement section of the online submission form. Please remove any funding-related text from the manuscript. 

4. We note that your Data Availability Statement is currently as follows: "All relevant data are within the manuscript and its Supporting Information files."

5. Please upload a new copy of Figure 1 as the detail is not clear. Please follow the link for more information: 

https://blogs.plos.org/plos/2019/06/looking-good-tips-for-creating-your-plos-figures-graphics/

https://blogs.plos.org/plos/2019/06/looking-good-tips-for-creating-your-plos-figures-graphics/

**Additional Editor Comments:**

This study provide additional reference for the early-stage assessment of FFR, which is an interesting topic. However it need further improvement. Please carefully check the reviewers' comments.

In addition, the following points should be considered.

1. There are some format errors (.Error! Bookmark not defined) in the tables.

2. In Table 1, I recommend to unify the digits of numbers.

3. FFRct is calculated based on CT images. Often the CT scan is recommended to the patients with symptoms. Therefore, regarding the one-year probability of MACE, the low value for FFTct need more discussion.

4. For the included literature, the inclusion and exclusion criteria need further clarification. The authors can consider a literature screening flowchart.

Reviewers' comments:

Reviewer's Responses to Questions

**Comments to the Author**

1. Is the manuscript technically sound, and do the data support the conclusions?

Reviewer #1: Yes

Reviewer #2: Yes

2. Has the statistical analysis been performed appropriately and rigorously? 

Reviewer #1: Yes

Reviewer #2: Yes

3. Have the authors made all data underlying the findings in their manuscript fully available?

Reviewer #1: Yes

Reviewer #2: Yes

4. Is the manuscript presented in an intelligible fashion and written in standard English?

Reviewer #1: Yes

Reviewer #2: Yes

5. Review Comments to the Author

Reviewer #1: This paper assessed the difference in healthcare costs and effects of fractional flow reserve derived from coronary computed tomography (FFRct) compared to standard diagnostics in patients with stable chest pain in the Netherlands. It is an excellent work. However, I have some suggestions.

1. Please provide more details about the decision-tree model.

2. What specific guiding value can this research provide to the clinic? Please discuss some suggestions for FFRCT technology in the future.

3. Figures are not clear, please provide high resolution pictures.

Reviewer #2: Good preliminary work of comparing the cost of HTA of FFRCTA against traditional diagnostic workflow of stable angina. Perhaps can include in discussion cost implications of this technology in resource limited environments/ expansion of this strategy to Asia Pacific (APAC) countries.

6. PLOS authors have the option to publish the peer review history of their article (what does this mean?). If published, this will include your full peer review and any attached files.

Reviewer #1: No

Reviewer #2: **Yes: **Han Bing Chow

---

## [Author Response · Author response to Decision Letter 0]

14 Feb 2024

Additional Editor Comments:

This study provide additional reference for the early-stage assessment of FFR, which is an interesting topic. However it need further improvement. Please carefully check the reviewers' comments.

In addition, the following points should be considered.

1. There are some format errors (.Error! Bookmark not defined) in the tables.

Thank you for this remark and for reviewing our manuscript. However, the version of Table 1 as shared with PlosOne does not show any format errors. Please see the screenshots below: 

2. In Table 1, I recommend to unify the digits of numbers.

We agree with the reviewer that it looks better when digits are unified. We unified the digits in Table 1 as much as possible, based on available data. 

Please see Table 1 on pages 8-9.

3. FFRct is calculated based on CT images. Often the CT scan is recommended to the patients with symptoms. Therefore, regarding the one-year probability of MACE, the low value for FFTct need more discussion.

Thank you for this comment. In the introduction it is explained that FFRct adds a mathematical simulation to the CT scan, and thereby increases the accuracy of the test. 

Please see page 5, lines 70-75:

To prevent unnecessary additional or invasive diagnostic procedures, measuring FFR from the non-invasive CCTA (FFRct) has been developed. With data from the CCTA scan, FFRct assesses the impact of stenosis in the coronaries using imaging data and mathematical simulations of the blood flow. Because of its accuracy, it could prevent unnecessary and costly treatments, including revascularizations, and diagnostic procedures without adding the burden and adverse events of an invasive technique [7-9]. 

Risk of MACE after an FFRct scan are lower than a CT scan due to the increased diagnostic accuracy of FFRct. To clarify this, it is added to the discussion section.

Please see page 15, lines 292-294:

Since FFRct adds a mathematical simulation to the traditional CCTA, thereby increasing the accuracy of the diagnostic test, it is always expected to lower the amount of MACE. 

4. For the included literature, the inclusion and exclusion criteria need further clarification. The authors can consider a literature screening flowchart.

Thank you for this comment. The inclusion and exclusion criteria used in the literature review are now added to the methods and results sections. 

Please see page 6, lines 99-106:

To collect data on costs and effects a literature search was performed in Pubmed using the following search terms: (((coronary artery disease[MeSH Terms]) AND (cardiac diagnostic method[MeSH Terms])) AND ((“2015/01/01”[Date – Completion] : “3000”[Date – Completion]))) AND ((coronary angiography[MeSH Terms]) OR (fractional flow reserve, myocardial[MeSH Terms]) OR (computed tomography[MeSH Terms])). Articles were excluded if they were published before 2015, if they did focus on other coronary malignancies than stable chest pain, and if they were not designed as a meta-analysis, systematic review or randomized controlled trial.

And page 12, lines 203-207:

Literature review

The literature review resulted in 197 results. After title and abstract screening, 162 articles were excluded. After full text review, twelve articles were selected for inclusion, see Figure 2.

Figure 2. Flow chart of literature review

Reviewer #1: This paper assessed the difference in healthcare costs and effects of fractional flow reserve derived from coronary computed tomography (FFRct) compared to standard diagnostics in patients with stable chest pain in the Netherlands. It is an excellent work. However, I have some suggestions.

1. Please provide more details about the decision-tree model.

The model is described in the methods section and illustrated in Figure 1. 

Please see pages 6-7, lines 97-128:

Model overview

A decision-tree model was developed that integrated data from published literature and information about unit costs to compare costs and clinical outcomes of FFRct compared to standard diagnostics. To collect data on costs and effects a literature search was performed in Pubmed using the following search terms: (((coronary artery disease[MeSH Terms]) AND (cardiac diagnostic method[MeSH Terms])) AND (("2015/01/01"[Date - Completion] : "3000"[Date - Completion]))) AND ((coronary angiography[MeSH Terms]) OR (fractional flow reserve, myocardial[MeSH Terms]) OR (computed tomography[MeSH Terms])). Articles were excluded if they were published before 2015, if they did focus on other coronary malignancies than stable chest pain, and if they were not a meta-analysis, systematic review or randomized controlled trial. The model simulates different diagnostic trajectories, including non-invasive anatomical imaging (i.e., CCTA), non-invasive functional imaging (i.e., magnetic resonance imaging (MRI)), and ICA with visual assessment and/or combined with a FFR measurement over a period of one year. The FFRct diagnostic strategy added the computational FFR analysis to the initial CCTA for all patients. Model outcomes included total costs from the hospital perspective, probability of correct diagnoses, and risk of major adverse cardiovascular events (MACE), defined as all-cause mortality, myocardial infarction (MI), and unplanned hospitalization for chest pain leading to urgent revascularization [8]. Outcomes were not discounted. The model was constructed in TreeAge Pro (version 2020; Williamstown, MA: TreeAge Software, Inc.). 

Patients with stable chest pain and an intermediate pre-test probability of obstructive CAD enter the model and received a CCTA (with or without FFR analysis depending on the diagnostic arm) (Figure 1). If the test findings were inconclusive, meaning the test provided insufficient information about the effect of the lesion on the blood flow (i.e., no positive or negative test result), patients will undergo non-invasive functional imaging (i.e., MRI). If those findings are also inconclusive, patients will undergo an ICA with visual assessment. Finally, when those test findings were inconclusive, patients received ICA with an FFR measurement, in line with the transmural agreements for patients with non-acute chest pain of the Dutch Association of Cardiologist [6, 11]. Patients who received a positive test result (either true or false) after a test were assumed to be treated with medications or a percutaneous coronary intervention (PCI) or coronary artery bypass grafting (CABG). Patients who received a negative test result (either true or false) receive no further treatment. Patients with a conclusive test result (positive or negative) were at risk of MACE.

It is unclear to us what more details about the decision tree-model need to be described.

2. What specific guiding value can this research provide to the clinic? Please discuss some suggestions for FFRCT technology in the future.

Thank you for this suggestion. We elaborated on the possible value this study could provide to a clinic in the discussion section. 

Please see page 17, lines 324-330:

For hospitals diagnosing patients with stable chest pain in The Netherlands, the results of this study indicate that the implementation of FFRct guided diagnostics would decrease the costs per patient, the amount of adverse events and increase the probability of correct diagnoses. For hospitals in the Netherlands considering to implement FFRct, this study shows promising results. Moreover, these study findings may contribute to the decision making process by the Dutch National Health Care Institute about reimbursement of FFRct as part of the standard health care package [31]. 

3. Figures are not clear, please provide high resolution pictures.

Thank you for noticing. We reviewed and re-uploaded all Figures and trust they are now in sufficient resolution.

Reviewer #2: Good preliminary work of comparing the cost of HTA of FFRCTA against traditional diagnostic workflow of stable angina. Perhaps can include in discussion cost implications of this technology in resource limited environments/ expansion of this strategy to Asia Pacific (APAC) countries.

Thank you for reviewing our manuscript and your kind words. Since our study focussed on the situation in The Netherlands, we currently do not have enough information to make statements about other situations. Nonetheless, we added a sentence to the discussion section to emphasize that different situations require a new analysis to be able to assess the potential value of FFRct. 

Please see page 17, lines 330-332:

The authors would like to emphasize that this study focused on the healthcare setting in The Netherlands, therefore other settings require an analysis tailored to those specific settings.

---

## [Decision Letter · Decision Letter 1]

16 Apr 2024

PONE-D-23-42875R1Early-stage health technology assessment of fractional flow reserve coronary computed tomography versus standard diagnostics in patients with stable chest pain in The NetherlandsPLOS ONE

Dear Dr. Boot,

Thank you for submitting your manuscript to PLOS ONE. After careful consideration, we feel that it has merit but does not fully meet PLOS ONE’s publication criteria as it currently stands. Therefore, we invite you to submit a revised version of the manuscript that addresses the points raised during the review process.

We look forward to receiving your revised manuscript.

Kind regards,

Haipeng Liu

Academic Editor

PLOS ONE

Journal Requirements:

Additional Editor Comments:

The majority of issues in previous comments have been addressed but there are some points for further improvement. Please check the comments and revise accordingly.

Reviewers' comments:

Reviewer's Responses to Questions

**Comments to the Author**

1. If the authors have adequately addressed your comments raised in a previous round of review and you feel that this manuscript is now acceptable for publication, you may indicate that here to bypass the “Comments to the Author” section, enter your conflict of interest statement in the “Confidential to Editor” section, and submit your "Accept" recommendation.

Reviewer #1: All comments have been addressed

Reviewer #3: All comments have been addressed

Reviewer #4: All comments have been addressed

2. Is the manuscript technically sound, and do the data support the conclusions?

Reviewer #1: Yes

Reviewer #3: Yes

Reviewer #4: Yes

3. Has the statistical analysis been performed appropriately and rigorously? 

Reviewer #1: Yes

Reviewer #3: Yes

Reviewer #4: Yes

4. Have the authors made all data underlying the findings in their manuscript fully available?

Reviewer #1: Yes

Reviewer #3: Yes

Reviewer #4: Yes

5. Is the manuscript presented in an intelligible fashion and written in standard English?

Reviewer #1: Yes

Reviewer #3: Yes

Reviewer #4: Yes

6. Review Comments to the Author

**Reviewer #1: **Overall, your study provides important evidence supporting the cost-effectiveness of FFRct compared to standard diagnostics in patients with stable chest pain. I commend you for your thorough analysis and contribution to the field. I look forward to seeing your manuscript published and its impact on clinical practice.

**Reviewer #3: **Table 1 shows a specificity of MRI higher than FFRct (94 vs 70). This needs to be mentioned and the advantages and disadvantages of both to be outlined in short. For example, the FFRct shows a better advantage in visualisation and MRI over quantification of capillary network or viability. The disadvantage could be higher contrast usage in FFRct.

**Reviewer #4: **This article presents an early-stage Health Technology Assessment (HTA) comparing Fractional Flow Reserve derived from coronary computed tomography (FFRct) to standard diagnostic methods for patients with stable chest pain in The Netherlands. The study aims to assess healthcare costs, the probability of correct diagnoses, and the risk of major adverse cardiovascular events (MACE) over one year. Utilizing a decision-tree model incorporating data from published literature, the analysis reveals that FFRct may potentially reduce overall healthcare spending and improve diagnostic accuracy and patient outcomes compared to standard diagnostics.

Key findings include:

1. The mean one-year healthcare cost per patient is lower for FFRct (€2680) compared to standard diagnostics (€2915).

2. FFRct demonstrates a higher one-year probability of correct diagnoses (0.78 vs. 0.61) and a lower risk of MACE (1.92x10^-5 vs. 0.01) compared to standard diagnostics.

3. Sensitivity analyses highlight the specificity of coronary computed tomography angiography (CCTA) and the probabilities and costs associated with percutaneous coronary intervention (PCI) as significant factors affecting the cost difference between FFRct and standard diagnostics.

4. The study suggests that for FFRct to be considered the least costly option, its added price should be below €935 per analysis.

5. The manuscript underscores the potential of FFRct to reduce unnecessary and costly treatments by enhancing the accuracy of non-invasive diagnostic procedures. It advocates for future cost-effectiveness studies to establish value-based pricing for FFRct and to quantify its economic value in therapeutic impact. The study contributes to the ongoing dialogue regarding the implementation of FFRct in clinical practice, emphasizing its potential benefits in the Dutch healthcare setting and suggesting further research and development, particularly in leveraging artificial intelligence for improved diagnostic predictions.

Here are some suggestions for improvement

1. Methodological Enhancements

1.1 Expand Data Sources: Incorporate a broader range of databases in your literature review, including EMBASE and Cochrane Library, to ensure a comprehensive understanding of the current research landscape.

1.2 Detailed Assumption Justification: Provide a more in-depth justification for each model assumption, including how they might bias results and potential methods for mitigating these biases.

2. Data and Analysis Improvements

2.1 Component Cost Analysis: Break down the cost analysis further to identify and discuss the specific components where FFRct offers the most significant cost savings over standard diagnostics.

2.2 Comparative Diagnostic Accuracy: Include a comparison of diagnostic accuracy rates between FFRct and other non-invasive methods across different studies to contextualize your findings.

2.3 Longitudinal Study Proposal: Propose a design for a longitudinal study to track the long-term outcomes of patients diagnosed using FFRct, including key metrics to be evaluated.

3. Practical Application and Policy Implications

3.1 Implementation Roadmap: Outline a detailed roadmap for the implementation of FFRct in clinical practice in The Netherlands, including steps to overcome potential barriers.

Guideline Integration: Suggest specific ways in which your findings could be integrated into national and international guidelines for CAD diagnosis.

3.2 Patient Stratification Criteria: Elaborate on the criteria for selecting patients for FFRct versus standard diagnostics to provide clearer guidance for clinicians.

3.3 AI Integration Strategy: Expand on the strategy for integrating AI into FFRct analysis, including potential challenges and solutions.

3.4 Study Recommendations for Policymakers: Offer detailed recommendations for future cost-effectiveness studies that policymakers could use to assess the value of including FFRct in reimbursement policies.

4. Expanding the Discussion

4.1 Broader Healthcare System Impact: Analyze the potential broader impacts of adopting FFRct on healthcare system efficiency, including the effects on hospital admission rates and overall patient management workflows.

4.2 International Benchmarking: Include a section comparing the adoption, outcomes, and cost-effectiveness of FFRct in The Netherlands with those in countries where it's more established, identifying lessons learned and best practices.

4.3 Reimbursement Policy Considerations: Dive deeper into how the study's findings could influence reimbursement policies, discussing potential hurdles and the economic implications for the healthcare system.

5. Enhancing the Research Framework

5.1 Sensitivity and Specificity Discussion: Beyond presenting the metrics, discuss their clinical significance and how they compare to the thresholds for clinical relevance in current practice.

5.2 Technology Adaptation Challenges: Discuss the technological, operational, and ethical challenges of adopting advanced AI techniques in FFRct analysis and how these can be addressed.

5.3 Future Research Directions: Clearly outline future research directions, including potential comparative studies with other diagnostic technologies and the exploration of FFRct's role in personalized medicine.

7. PLOS authors have the option to publish the peer review history of their article (what does this mean?). If published, this will include your full peer review and any attached files.

Reviewer #1: No

Reviewer #3: **Yes: **Mark Christopher Arokiaraj

Reviewer #4: No

---

## [Author Response · Author response to Decision Letter 1]

15 May 2024

Please see the file "Response to reviewers_20240513"

Reviewer #1: Overall, your study provides important evidence supporting the cost-effectiveness of FFRct compared to standard diagnostics in patients with stable chest pain. I commend you for your thorough analysis and contribution to the field. I look forward to seeing your manuscript published and its impact on clinical practice.

We appreciate the reviewer’s effort to review the manuscript and thank the reviewer for the kind words.

Reviewer #3: Table 1 shows a specificity of MRI higher than FFRct (94 vs 70). This needs to be mentioned and the advantages and disadvantages of both to be outlined in short. For example, the FFRct shows a better advantage in visualisation and MRI over quantification of capillary network or viability. The disadvantage could be higher contrast usage in FFRct.

To address the reviewer’s comment we added information to the discussion. See page 17, lines 328-331:

Besides, it is noteworthy that the specificity of MRI is higher than FFRct, see Table 1. Although it is expected that the combination of FFRct with CCTA would significantly improve diagnostic specificity [7], future studies could also give more clarity.

Reviewer #4: This article presents an early-stage Health Technology Assessment (HTA) comparing Fractional Flow Reserve derived from coronary computed tomography (FFRct) to standard diagnostic methods for patients with stable chest pain in The Netherlands. The study aims to assess healthcare costs, the probability of correct diagnoses, and the risk of major adverse cardiovascular events (MACE) over one year. Utilizing a decision-tree model incorporating data from published literature, the analysis reveals that FFRct may potentially reduce overall healthcare spending and improve diagnostic accuracy and patient outcomes compared to standard diagnostics.

Key findings include:

1. The mean one-year healthcare cost per patient is lower for FFRct (€2680) compared to standard diagnostics (€2915).

2. FFRct demonstrates a higher one-year probability of correct diagnoses (0.78 vs. 0.61) and a lower risk of MACE (1.92x10^-5 vs. 0.01) compared to standard diagnostics.

3. Sensitivity analyses highlight the specificity of coronary computed tomography angiography (CCTA) and the probabilities and costs associated with percutaneous coronary intervention (PCI) as significant factors affecting the cost difference between FFRct and standard diagnostics.

4. The study suggests that for FFRct to be considered the least costly option, its added price should be below €935 per analysis.

5. The manuscript underscores the potential of FFRct to reduce unnecessary and costly treatments by enhancing the accuracy of non-invasive diagnostic procedures. It advocates for future cost-effectiveness studies to establish value-based pricing for FFRct and to quantify its economic value in therapeutic impact. The study contributes to the ongoing dialogue regarding the implementation of FFRct in clinical practice, emphasizing its potential benefits in the Dutch healthcare setting and suggesting further research and development, particularly in leveraging artificial intelligence for improved diagnostic predictions.

Here are some suggestions for improvement

1. Methodological Enhancements

1.1 Expand Data Sources: Incorporate a broader range of databases in your literature review, including EMBASE and Cochrane Library, to ensure a comprehensive understanding of the current research landscape.

As suggested by the reviewer, we searched the databases EMBASE and Cochrane Library for FFRct studies which reported on similar outcomes as we focussed on in our study. The EMBASE library did not result in any new insights, which might be explained by its biomedical and pharmacological focus. Based on results from our search in the Cochrane Library we added information of two studies to the introduction. See page 6, lines 87-91:

A study performed in the United States showed similar costs for the initial evaluation of chest pain using FFRct compared to standard care pathways (cost difference +7% (+$324), CI -12% to +26%, p = 0.49) [11]. A single-centre study performed in China also indicated that FFRct strategy is associated with a lower 1-year risk of MACE when compared to CCTA [12].

1.2 Detailed Assumption Justification: Provide a more in-depth justification for each model assumption, including how they might bias results and potential methods for mitigating these biases.

We are aware that performing an early HTA study requires to make assumptions, which may not always perfectly reflect reality However, assumptions here should be regarded as educated guesses and as such useful to support decision making. Notwithstanding, to not ignore the use of assumptions, we added the following text to the discussion. See page 16, lines 310-320:

In our study a few assumptions were made. First, it was assumed that the diagnostic accuracy of ICA with FFR is one. If this accuracy would be lower in reality, this is expected to increase the risk of MACE in both treatment arms, but more in the standard of care arm since more people undergo invasive testing in that arm. Second, assumptions were made regarding the percentage of inconclusive test results. If this percentage would be higher in reality, both treatment arms would perform more diagnostic tests, both invasive and non-invasive. The opposite if the actual percentage would be lower. The first is expected to result in a greater difference between the two strategies, and the second in a smaller difference. At last, assumptions were made regarding drug prescription. The results of the sensitivity analyses show that costs of medications only have a small impact on the outcomes, see Figure 3. 

2. Data and Analysis Improvements

2.1 Component Cost Analysis: Break down the cost analysis further to identify and discuss the specific components where FFRct offers the most significant cost savings over standard diagnostics.

For each treatment strategy we calculated the costs in detail. Please see Appendix A, Table 1. Here, the (main) contributors to the costs of each treatment strategy can be seen. We believe that incorporating this level of detail in the sensitivity analysis rather confuses than clarifies the analyses. Hence, Appendix A is offered to readers.

2.2 Comparative Diagnostic Accuracy: Include a comparison of diagnostic accuracy rates between FFRct and other non-invasive methods across different studies to contextualize your findings.

The aim of this early-HTA study is to analyse healthcare costs and effects of FFRct guided diagnostic pathway compared to standard diagnostics. Differences in outcomes are a result of the different accuracies and characteristics of the diagnostic tools. These main outcomes are discussed in the discussion. 

2.3 Longitudinal Study Proposal: Propose a design for a longitudinal study to track the long-term outcomes of patients diagnosed using FFRct, including key metrics to be evaluated.

We thank the reviewer for this suggestion and added the following lines to the discussion, please see page 18 lines 354-356:

Besides, if FFRct would be implemented, a longitudinal study is suggested to collect additional data on costs, diagnostic accuracy, and risk of MACE.

3. Practical Application and Policy Implications

3.1 Implementation Roadmap: Outline a detailed roadmap for the implementation of FFRct in clinical practice in The Netherlands, including steps to overcome potential barriers.

Guideline Integration: Suggest specific ways in which your findings could be integrated into national and international guidelines for CAD diagnosis.

We appreciate the reviewer’s thought about the implementation plan of the new technique. The goal of the early HTA study is to inform decision and policy makers about the potential value of the new technique. If it is decided to implement the technique in the healthcare system, an implementation roadmap would indeed be very helpful. However, we believe that it is not valuable to add to this manuscript at this stage of the research. 

3.2 Patient Stratification Criteria: Elaborate on the criteria for selecting patients for FFRct versus standard diagnostics to provide clearer guidance for clinicians.

At this moment, sufficient information to state elaborate statements about this issue is lacking. However, the expected effects of implementing FFRct among different patient groups is discussed in the discussion section. Please see page 15, lines 288-294:

People with a higher pre-test probability are more likely to be directly referred to invasive tests. In this case we would expect the differences in costs to increase in favour of FFRct guided diagnostics, the amount of correct diagnosis to increase and the risk of MACE to decrease. People with a lower pre-test probability usually first receive behavioural change advices. If this is not effective, they will usually undergo diagnostic imaging, starting from CCTA. Therefore, no differences in outcomes are expected compared to people with an intermediate pre-test probability when people are referred to imaging.

3.3 AI Integration Strategy: Expand on the strategy for integrating AI into FFRct analysis, including potential challenges and solutions.

The aim of this study was to examine whether FFRct in its current form has the potential to be a cost-effective diagnostic tool in the Dutch healthcare setting. Although we agree with the reviewer that more information on the integration of AI in the analysis would be interesting, we believe that this requires more research and should be presented in a separate manuscript in the future.

3.4 Study Recommendations for Policymakers: Offer detailed recommendations for future cost-effectiveness studies that policymakers could use to assess the value of including FFRct in reimbursement policies.

As mentioned before, this early-HTA study is performed to provide an assessment of the potential value of a new technique at an early phase. Hence, it is mentioned in the last paragraph of the manuscript, that future cost-effectiveness studies are needed to quantify the economic value of the therapeutic impact. See page 18, lines 353-354:

Future cost-effectiveness studies should determine a value-based pricing for FFRct and quantify the economic value of the anticipated therapeutic impact.

4. Expanding the Discussion

4.1 Broader Healthcare System Impact: Analyze the potential broader impacts of adopting FFRct on healthcare system efficiency, including the effects on hospital admission rates and overall patient management workflows.

We acknowledge that it would be interesting to also evaluate the broader impact on the healthcare system. To emphasize this, text is added to the discussion. See page 18, lines 356-358:

Future studies could also focus on the broader healthcare system impact, including the impact of the implementation of FFRct guided diagnostics on healthcare system efficiency.

4.2 International Benchmarking: Include a section comparing the adoption, outcomes, and cost-effectiveness of FFRct in The Netherlands with those in countries where it's more established, identifying lessons learned and best practices.

Regarding our information, there is no western country where the FFRct technique is already implemented as standard of care. 

The potential value of FFRct has been researched in the UK. Results of these studies are discussed in the manuscript. See page 15, lines 271-283:

The study findings are consistent with the NICE recommendations which indicate that FFRct-guided diagnostics may lead to cost savings [10]. Also, results from the PRECISE randomized trial endorse the results of more correct diagnoses by showing that more accurate catheterizations are performed with FFRct testing, without increasing the risk of MACE [21]. At last, additionally an increased catheterization lab efficiency and a reduction in additional non-invasive testing was shown in England in the FISH&CHIPS study [22]. However, individual trials do suggest, albeit non-significantly, higher costs in the FFRct-guided strategy compared to the usual care after planned non-invasive testing [8, 23]. In the FORECAST trial, for example, a multi-centre trial with 1,400 patients with stable chest pain were randomized to either CCTA with selective FFRct or standard care, which were dependent on their pre-test probability of CAD [23], showed a non-significantly higher costs in the FFRct-guided strategy compared to the usual care group at 9 months (£1,605 versus £1,491; p=0.10). Differences in results can be explained by differences in standard diagnostic pathways between countries. 

4.3 Reimbursement Policy Considerations: Dive deeper into how the study's findings could influence reimbursement policies, discussing potential hurdles and the economic implications for the healthcare system.

We appreciate the reviewer’s considerations. As mentioned in the introduction, favourable findings in this study could justify further investments in research and development of the technology. Since the results of this study are indeed favourable for FFRct, investments in the technology can be considered by policy makers. Awaiting those decisions, no information about expected hurdles and economic implications is available yet.

5. Enhancing the Research Framework

5.1 Sensitivity and Specificity Discussion: Beyond presenting the metrics, discuss their clinical significance and how they compare to the thresholds for clinical relevance in current practice.

Considering the aim of this manuscript, i.e. to present the findings from an early-HTA study of FFRct in the Netherlands. As part of this, we present the sensitivity and specificity of tests. In addition, we present variables regarding the costs of tests. Next, comparisons are made between tests regarding the incremental cost-effectiveness and conclusions are drawn from this. Hence, our discussion is focused on the incremental cost-effectiveness of tests which is exactly what may be expected from an early-HTA study. For discussions about the clinical significance, multiple references with detailed information are included (7, 9, 12 and 13).

5.2 Technology Adaptation Challenges: Discuss the technological, operational, and ethical challenges of adopting advanced AI techniques in FFRct analysis and how these can be addressed.

The aim of this manuscript is to present the results of an early-HTA study of the FFRct technique in its current state, compared to the standard of care in the Netherlands. At this moment, AI is not part of the technique, and therefore not discussed in this manuscript.

5.3 Future Research Directions: Clearly outline future research directions, including potential comparative studies with other diagnostic technologies and the exploration of FFRct's role in personalized medicine.

In the manuscript, a few future research studies are recommended in this respect. See page 18, line 353-358:

Future cost-effectiveness studies should determine a value-based pricing for FFRct and quantify the economic value of the anticipated therapeutic impact. Besides, if FFRct would be implemented, a longitudinal study is suggested to collect data on costs, diagnostic accuracy, and risk of MACE. Future studies could also focus on the broader healthcare system impact, such as the impact of the implementation of FFRct guided diagnostics on healthcare system efficiency.

And page 15 Line 294-296: 

For conclusive findings regarding costs, additional analyses could be performed to compare different pathways to FFRct guided diagnostics.

---

## [Decision Letter · Decision Letter 2]

27 May 2024

Early-stage health technology assessment of fractional flow reserve coronary computed tomography versus standard diagnostics in patients with stable chest pain in The Netherlands

PONE-D-23-42875R2

Dear Dr. Boot,

We’re pleased to inform you that your manuscript has been judged scientifically suitable for publication and will be formally accepted for publication once it meets all outstanding technical requirements.

Kind regards,

Haipeng Liu

Academic Editor

PLOS ONE

Additional Editor Comments (optional):

Reviewers' comments:

Reviewer's Responses to Questions

**Comments to the Author**

1. If the authors have adequately addressed your comments raised in a previous round of review and you feel that this manuscript is now acceptable for publication, you may indicate that here to bypass the “Comments to the Author” section, enter your conflict of interest statement in the “Confidential to Editor” section, and submit your "Accept" recommendation.

Reviewer #3: All comments have been addressed

2. Is the manuscript technically sound, and do the data support the conclusions?

Reviewer #3: Yes

3. Has the statistical analysis been performed appropriately and rigorously? 

Reviewer #3: Yes

4. Have the authors made all data underlying the findings in their manuscript fully available?

Reviewer #3: Yes

5. Is the manuscript presented in an intelligible fashion and written in standard English?

Reviewer #3: Yes

6. Review Comments to the Author

Reviewer #3: Further large multicentric studies are required. The concept is promising, and can be included in screening CT's for early and effective evaluation of coronary artery disease.

7. PLOS authors have the option to publish the peer review history of their article (what does this mean?). If published, this will include your full peer review and any attached files.

Reviewer #3: **Yes: **Mark Christopher Arokiaraj

---

## [Editor Report · Acceptance letter]

3 Jun 2024

PONE-D-23-42875R2 

PLOS ONE

Dear Dr. Boot, 

I'm pleased to inform you that your manuscript has been deemed suitable for publication in PLOS ONE. Congratulations! Your manuscript is now being handed over to our production team.

Kind regards, 

on behalf of

Dr. Haipeng Liu 

Academic Editor

PLOS ONE